# Finding One Missing Puzzle of Contextual Word Embedding: Representing Contexts as Manifold

## Abstract

The current understanding of contextualized word embedding interprets the representation by associating each token to a vector that is dynamically modulated by the context. However, this "token-centric" understanding does not explain how a model represents context, leading to a lack of characterization from such a perspective. In this work, to establish a rigorous definition of "context representation", we formalize this intuition using a category theory framework, which indicates the necessity of including the information from not only individual contexts but also how transitions happen among the semantic effect of different contexts. As a practical instantiation of our theoretical understanding, we also show how to leverage manifold learning methods to characterize how a representation model (i.e., BERT) encodes different contexts and how a representation of context changes when going through different model components such as attention and FFN. We hope this novel theoretical perspective sheds light on the further understanding of Transformer-based language representation models.

## 1 Introduction

In modern natural language processing, using vector representation of words in a low dimension space is a common practice. In early days, word embeddings were static, i.e., the representation is solely determined by a word's identity, such as the case of word2vec (Mikolov et al., 2013) and GloVe (Pennington et al., 2014). In contrast, contextualized word embeddings, e.g., ELMo (Peters et al., 2018), BERT (Devlin et al., 2019) and GPT (Brown et al., 2020), revolutionized this technique by introducing information from the contexts to the central word.

When compared to static embedding, studies to understand the contextual embedding generally lag behind. This is particularly unfavoured as the superior performance of pre-trained contextualized representations is gaining more and more attention. To tackle this problem, previous studies have proposed several directions to explore. For example, researchers in Hewitt & Manning (2019); Coenen et al. (2019); Wu et al. (2020) develop several probing techniques to relate contextualized word embedding to syntactic features. On the other hand, several works try to uncover how contexts influence the geometry of embedding space. Ethayarajh (2019) measures the context-specificity of the word embeddings from different layers of several deep representation models and confirms the majority variance is provided by the context but not the token. They also point out that the embedding space is generally anisotropic. Then, results from Cai et al. (2021) indicate that isotropy actually exists in certain isolated clusters of words in the representation space. Mamou et al. (2020) proposes to analyze the manifolds defined by word identity and other linguistic features. Under both "predictive" (i.e., with [MASK] tokens) and "contextualized" (i.e., the real sentence) settings, the authors analyze how well the manifolds can be separated from each other.

While these pioneering works offer some useful insight, their starting points are generally "token-centric", i.e., the analysis focuses on how the context affects the representation of tokens. Therefore, very few studies touch upon another critical problem, i.e., for contextualized word embedding methods, how exactly does the model represent the context itself. In this study, to fill in this gap, we introduce a novel perspective to analyze the representation of context. To give a rigorous discussion, we first provide a set of mathematical definitions by formulating the language representation prob-

lem using category theory. Then, inspired by manifold learning, we provide two analysis scenarios and also methods to probe the representation of contexts. As a realization of these theoretical analyses, we further provide empirical results and cases studies, which give more insights into the current contextual representation models. A particularly interesting example can be the re-thinking of the functions of Multi-head attention and Feed Forward Network (FFN) in a Transform-based model.

## 2 MOTIVATION AND CONTRIBUTIONS

This work tackles an untouched fundamental problem in language representation learning, i.e., how a representation model represent different contexts. Therefore, it provides a novel perspective to re-think and characterize these models.

In a contextualized representation model, the representation at a specific sequence position is determined by both the token at that place and the surrounding context. However, most of the attention has been paid to one direction, i.e., understanding how the vector represents the token, leaving the context part largely unexplored. Intuitively, we argue that the effect of different context conditions (termed as observed condition below) should be semantically related. This point, however, is not explicitly modeled in the current representation scheme, which motivates us to develop this study.

Ideally, a representation scheme of context should have the following two features. Firstly, it should be compatible with the word representation mentioned above. Secondly, the effect of the central token and observed condition should be disentangled, which makes the latter focusing on how different they are in affecting the central word's semantics. To achieve these goals, we design the following representational scheme. Firstly, for each observed condition, we fix the context, mutate the central token, and record how much is changed about the representation vector of the central token. This generates a collection of vectors whose each element corresponds to a contextualized representation. We hope this kind of collection reflects how each observed condition affects the semantics of the different central tokens, and comparing different collections reflect how different contexts affect the semantics. On the other hand, as mentioned above, to disentangle the effect of observed condition and central token, we study the nature of the collection, other than its elements.

However, characterizing this kind of information is challenging. Obviously, taking the collection of mutated sequences (and the associated vectors) as a set would miss a lot of information among them. Therefore, we propose a systematic and rigorous mathematical formulation based on category theory which states clearly the representation of context should reflect both the set of representations of the substituted central tokens as well as how these sets of word representations can transit to each other. In particular, we introduce the concept "morphism" to characterize the mappings between different token substitution sets (corresponding to the semantic effects of different observed conditions). Furthermore, with of help of manifold learning, we also develop a tool set to analyze the aforementioned information experimentally, from either a functional or a topological perspective.

Besides the above contributions, our empirical results, enabled by these theoretical foundations, also generate several interesting findings. First, our analysis of context around [CLS] token suggests that high-confidence correct predictions tend to occur when contexts show robustness in modulating the semantics of the central token as well as consistency of the modulating effect across different samples. Last but not least, our dissection of the attention and FFN layers identifies a game between them when information gets through a deep model, and their final agreement actually supports our hypothesis that they both of them try to fit an identity morphism that reflects *bona fide* linguistic associations.

## 3 PRELIMINARY: CONTEXTUAL LANGUAGE REPRESENTATION

Following the distributional hypothesis (Harris, 1954), the output vector of a representational model (denoted by $G$) represents the probability of the appearance of certain tokens (i.e., context) around a given token (i.e., central token $x$), i.e., $p(context|x)$. Note that here context is a random variable, whose formal definition will be given in the next section.

In contextualized language modeling (Peters et al., 2018), the distribution of context is conditioned not only on the central token but also on what really appears around the central token in each sample (e.g., the unmasked tokens in a given sentence). In this sense, we introduce observed condition $C$ to

denote the observed context. That is to say, observed condition C corresponds to an actual value of random variable context given a specific sample, which makes the modeling target of contextualized language modeling be $p(context|C, x)$.

In this work, we propose to represent each observed condition $C$ by inspecting how the model represents various central tokens given this condition, which gives the following collection of vector, i.e.,

$$\{G(C, x)|x \in Vocab\} \tag{1}$$

where $Vocab$ is the set of distinct tokens used by he tokenizer. However, the definition of set only provides rough description of the nature of contexts, which motivates as to introduce category for a rigorous definition, as described below.

## 4 CATEGORY THEORY FORMULATION AND NOTATIONS

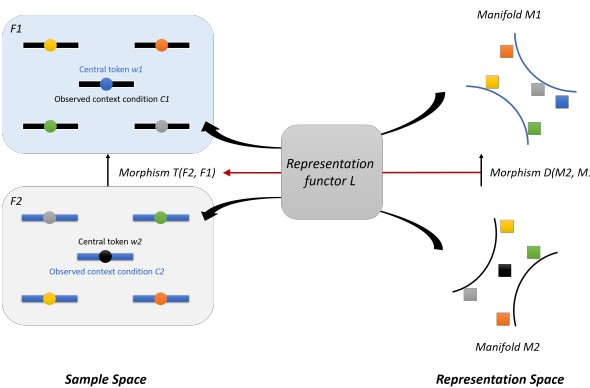

Figure 1: Illustration of the proposed representation scheme. For each observed condition, we propose to characterize its semantic modulation effect by a collection $F$ of sentences with different the central tokens. Then we use category theory to formulate the properties of this collection. As each substituted sequence can be represented by a language model, we can also formulate a category definition for vectors, which can be further characterized by manifold analysis. Therefore, we can use the properties of manifold $M$ to represent the properties of collection $F$.

As mentioned in Section 2, in our problem formulation, "context" is a random variable. So here we first give the encoding of each observed condition (i.e., an actually appearing context) using the formal language of probability theory (Stroock, 2010). Specifically, in BERT-style language model, we use a three-dimensional tensor (corresponding to token species, position and type, respectively) to define the sample space and take the $\sigma$-field of the sample space as the event space. Note that to describe the collection of probability distribution of different central tokens, we introduce a discrete random variable set $K = \{k_i\}$ and the distribution of each $k_i$'s image corresponds to the distribution of context given a certain central token $i$ (Note that $i$ is also defined as a triplet of species, position and type). Concretely, we have:

**Definition 4.1**. Let $K$ be a finite discrete set whose each element $k_i : \Omega \to E$ is a random variable that projects from the discrete sample space of tokens $\Omega \in \mathbb{N}^N$ to its $\sigma$-field $E$, where $N$ is set to three, corresponding to token species, token type and token position, respectively. The cardinal number of $K$ is marked as $n$.

Therefore, each element $k_i$ indicates the presence of a certain token (with defined species, type and position) as the central token that accompanies the context. Following the principle of distributional hypothesis, the representation of each central token represents the a probability density function of the context, which gives:

**Definition 4.2** An observed condition $C$ is encoded as a set $F(C)$ whose each element can be expressed as a probability density function $f_{k_i}$ conditioning on the observed condition $C$, i.e., $F(C) = \{f_{k_i}(context = e|C, context \in E)|i \in \Omega\}$.

Ultimately, one expects the representation of observed conditions to reveal a comprehensive characterization that transfers well to a general scope of downstream applications. In particular, the following two features are considered here, 1) for each observed condition, how it can affect the semantics of the central token, and 2) how different observed conditions modulate the semantics differently. This intuition is reminiscent of category theory that explicitly study the *objects* and their mappings (i.e.,*morphisms*).

**Definition 4.3** (Category of observed conditions) To define the objects in a category of observed conditions, we have

$$ob(\mathbb{F}) = \{F(C) | C \in E\} \tag{2}$$

which contains the probability density of all possible context events. Naturally, we can define an abstract collection of morphisms that map from one object to another, i.e.,

$$hom(\mathbb{F}) = \{hom(F_i, F_j) | F_i, F_j \in ob(\mathbb{F})\}, \tag{3}$$

where

$$hom(F_i, F_j) \triangleq \{T_{ij} : F_i \to F_j | T_{ij}(f_k(X = e | C_i, e \in E)) = f_k(X = e | C_j, e \in E)\}, \forall k \in K. \tag{4}$$

Note that through the above definition, we introduce several constrains that reflect the nature of language. First, for each set of morphism between two objects, there only exists one morphism, assuming that the model should learn to capture one genuine relationship between two observed conditions. Second, this morphism is a bijection defined on $F$ (note the subscript $k$ in Eq (4)), so each probability density of context should be projected to a unique partner with the same central token and vice versa (i.e., this morphism would not cause semantic collision when central tokens are different). We will discuss how our experimental results support these constrains later.

After the definition of categories of observed conditions, we move on to its representation. Note that each observed condition corresponds to a token substitution set (Definition 4.2), and a model $G$ is trained to perform injection from each probability density function in the set to a representation vector, i.e., $G(C) : F \to M$, where $M$ consists of vectors in a $d$-dimensional Euclidean space. Therefore, we can similarly define the category of vectors as the representation of category of observed conditions. What is different here is we how have to take different representation models into consideration, which gives the following definition:

**Definition 4.4** (Category of vector representations) Suppose we have a set of $G$ models: $G_{set} = \{G_m\}$, the objects of category of vector representations can be defined as

$$ob(\mathbb{M}) = \{M_{C_i G_m}\}, \tag{5}$$

where

$$M_{C_i G_m} = \{V_k | k \in K, G_m \in G_{set}, V_k \triangleq G_m(f_k(X = e | C_i, e \in E))\}, \tag{6}$$

The definition of morphism collections also considers the difference in model, which gives:

$$hom(\mathbb{M}) = \{hom(M_{C_i G_m}, M_{C_j G_n}) | M_{C_i G_m}, M_{C_j G_n} \in ob(\mathbb{M})\} \tag{7}$$

where

$$hom(M_{C_i G_m}, M_{C_j G_n}) = D_{C_i G_m \to C_j G_n} : M_{C_i G_m} \to M_{C_j G_n}, \tag{8}$$

$$D_{C_i G_m \to C_j G_n}(G_m(F(C_i))) = G_n(F(C_j)), \tag{9}$$

and $C_i$ and $C_j$ stand for two different observed conditions. Given the above definitions, we can then formulate the representation of context by a functor $L$ from this category $\mathbb{M}$ to category $\mathbb{F}$, i.e.,

$$L(G_m(C_i)) = F(C_i), \tag{10}$$

$$L(D_{C_i G_m \to C_j G_n}) = T_{ij}, \forall D_{C_i G_m \to C_j G_n} \in hom(M_{C_i G_m}, M_{C_j G_n}), \tag{11}$$

where $G_m, G_n \in G_{set}$.

## 5 UNDERSTANDING REPRESENTATION OF CONTEXT UNDER THE CATEGORICAL FRAMEWORK

In this section, we will explain how to analyze the representation of category $\mathbb{F}$ with given by category $\mathbb{M}$ output by the word representation model $G$.

## 5.1 Representing objects of category $\mathbb{F}$

Given a certain observed condition, Eq.(10) defines a *object-to-object* mapping of the vector representations given by $G$ to a set $F$ of probability density functions. Therefore, the nature of each object in category $\mathbb{F}$ can be analyzed by converting each sequence in the token substitution set to a vector, which is consistent with the common usage of word representation models. Note that the definition of $\mathbb{M}$ does not require model $G$ to be determined, so our formulation includes a collection of representations potentially given by different $G$.

## 5.2 Representing morphisms of category $\mathbb{F}$

Apart from the "token-centric" understanding of contextual language representation, this work emphasizes on the concept that the learned functor $L$ should also represent the morphisms (each morphism is corresponds to a transition from one object to another) from category $\mathbb{M}$ to category $\mathbb{F}$, i.e., the mapping of *inter-object* structure between the two categories. More specifically, the definition given by Eq.(4) and Eq.(7) indicates both the learned model $G$ and observed condition $C$ play a role in the representation of morphisms. Here, we propose to analyze the two factors side-by-side in the following two scenarios. **Scenario A.** The representation of different observed conditions when the representation model $G$ is fixed ($m = n$ in Eq.(8)). This scenario characterizes how the morphism between any two objects in category $\mathbb{M}$ (i.e., two collections of representation vectors) represent their counterparts in category $\mathbb{F}$, with the help of the learned representation model $G$. This scenario reflects how the model distinguishes different observed conditions in modulating the semantics of central tokens, by taking into account that this difference should be represented by some relationship of the whole collection of representation vectors (not any single vector alone). **Scenario B.** The representation of a certain observed condition $C$ given different representation models. This scenario extends the understanding of the representation model from a static view (i.e., the model parameters are already learned), to a dynamic view (i.e., the model can change either in configuration or parameter). This gives more implication of our theoretical perspective in dissecting the model components or the learning process. In this work, we select the first angle. By treating the output given by each layer of a Transformers model as outputs from different $G$ models, we can actually gain more insights into how the representation of context evolves across the different layers of a pre-trained language model. Considering the constraints introduced in **Definition 4.3**, different models should all attempt to represent the same identity morphism, i.e., $id_{T:F \to F}$. This makes sense in correspondence with the following semantics understanding: The relationship of semantic functions between two observed conditions should be determined by the language itself but not the model. Keeping this mind, we can measure the consistency of morphism representation across each layer in a pre-trained model as a probe for the model's behavior.

## 6 Evaluating changes in representation of context: a manifold perspective

The aforementioned reasoning indicates the necessity of characterizing the representation of morphisms as a complement to the characterization of representation vectors alone. Therefore, here we focuses on how morphisms in category $\mathbb{M}$ reflect the nature of category $\mathbb{F}$. More specifically, the elements of the manifolds are obtained following the protocol below, in the spirit of distributional hypothesis: Given a central word $x$ appearing with a real-world context $C$, we substitute $x$ with all possible tokens in the vocabulary of the language model (Note that for the sake of computation efficiency, this sampling strategy fixes the dimension of position embedding as in the real context). The collection of sampled elements is denoted as a token substitution set. To facilitate the characterization of structural information, following the manifold assumption (Jolliffe, 2011), we hypothesize that the vector representation of all the probability distributions corresponding to different central tokens lie on a low-dimensional manifold in the $d$-dimensional Euclidean space. With this approximation, while the exact solution of morphisms $D$ is still unsolved, we can leverage comparison of manifold properties to study the behavior of $D$ empirically, i.e.,

$$D^* : \mathbb{R}^d \times \mathbb{R}^d \to \mathbb{R}^{n \times n} = Comparison(Q(D(G(\circ))), Q(G(\circ))) \tag{12}$$

where $D^*$ is the change in manifold property that is induced by $D$, $Q$ is certain characterization of the manifold. In this work, we propose two specific forms of $Q$: (1) a task-related projection of the manifold, and (2) the topology output by a manifold learning algorithm.

**Functional characterization**  When comparing two contexts, a straightforward way is to project each representation vector to a task-related score. In practice, one can choose a linear head that is either pre-trained along with the representation model, or fine-tuned for a specific downstream task. Then we can use statistics of the task-related scores to analyze the behaviour of each manifold.

**Topological characterization**  To further characterize the topology of the proposed manifold, we can also leverage the toolset from manifold learning community to reveal the local features. In practice, we perform an inspection of the existence of topological associations in the local region. Particularly, here we borrow the method for topology determination from a prominent manifold learning algorithm, i.e., UMAP (McInnes & Healy, 2018). The local topology is determined among k-nearest neighbors, and represented by a fuzzy set to indicate whether an edge between nodes exists. Please refer to the original UMAP paper for a more detailed description. Note that as UMAP does not offer an estimation in global geodesic distances for arbitrary two elements (it focuses on local topology), our analysis also does not include this part. Nevertheless, we show below that the local information alone can provide some interesting insights for language modeling.

## 7 EXPERIMENTS

### 7.1 MODELS AND SETTINGS

In this study, to provide insights originating from our new understanding, we analyze the word representation of a widely used contextualized embedding model, i.e., BERT (Devlin et al., 2019). The implementation, feature extraction and pre-trained model ("Bert-base-uncased") are all based on the Transformers package [1].

Our analysis involves two datasets. Wikitext-2 (Merity et al., 2017) is used as a representative of pre-training corpus, while SST-2 (Socher et al., 2013), one of the GLUE dataset (Wang et al., 2018), is used as a typical downstream task (i.e., sentiment analysis of movie reviews) dataset. Detailed statistics of data used in the analysis will be provided in the corresponding sections. To fine-tune the model for the SST-2 task, we use a learning rate of $2e^{-5}$, batch size of 32, and epoch number of 3. For the topology analysis using UMAP, we keep the original hyperparameter of kNN as 15.

In the two subsections below, we provide some empirical results corresponding to the two scenarios defined in Section 5.

### 7.2 UNDERSTANDING DIFFERENT REPRESENTATION OF CONTEXTS FROM A PRE-TRAINED MODEL

**Star graph as a unit of analysis.**  To give a straightforward impression of the topology of a context manifold, here we first list an example (Figure 2(a)). The example context comes from Wikitext-2. Here, we visualize the connections of the real context to its neighbors, as determined by the local fuzzy set as in UMAP, using the output of the BERT final layer. Note that each context here is associated with a set of substitutions of the original token to other tokens. By treating the original sentence as the center, the topological evaluation can be conducted by examining the star-graph. Note that while our analysis of topology only involves this star graph (i.e., omitting other potential connections), the results in Section 7.3 still reveal some interesting insights into the representational model. We leave the exploration of more complicated topology for future work.

**How contexts modulate the semantics: a case study of [CLS].**  To provide a more task-related perspective, we also study how contexts are represented for the special token [CLS], whose representation is often used as input for natural language classification. Intuitively, as [CLS] itself has little semantics, the linguistic feature of the whose sentence should all come from each observed condition. In correspondence to our formulation, the token substitution set's representations are

---
[1]https://huggingface.co/transformers/

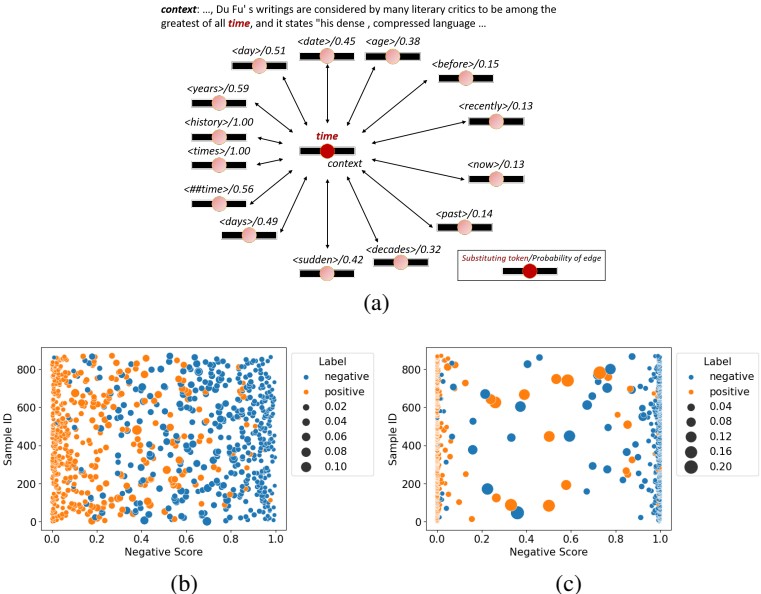

Figure 2: (a) An example of star graph where each element in the token substitution set has a probability of connecting to the real context, as determined by the local fuzzy set as in UMAP. (b) The effect of topology of [CLS] token before fine-tuning, as evaluated by functional characterization. (c) The effect of topology of [CLS] token after fine-tuning, as evaluated by functional characterization. For (b) and (c), the radius of a dot denotes the standard deviation caused by local topology. The larger the radius is, the larger the standard deviation will be.

projected, together with the [CLS] token's representation itself, by the linear classification layer that is learned during fine-tuning. Given an input (original or substituted), the linear layer serves as a projection from the last representational layer to a scalar value to indicate the classification result.

In light of this understanding, we plot the classification score (for the negative class) of each sample (denoted by the dots), together with the standard deviation (denoted by the radius of dots) of scores when using the substituted contexts as input. The results from pre-trained (a linear classifier on the top of frozen BERT encoder) and fine-tuned representation model is plot in Figure 2(b) and Figure 2(c), respectively. In this experiment, we leverage two features of a sample to conduct the analysis, i.e., the classification score of the original sentence beginning with [CLS] and the standard deviation of classification scores of all the elements in the token substitution set. In particular, the classification score indicates the task-related (a binary sentiment analysis) semantics of the sample. And the standard deviation represents the task-related magnitude of semantic shifts across the token substitution set, which simplifies the comparison between different sets to the comparison of different standard deviation values. We then show how we analyze the representation of objects and morphisms of context category $\mathbb{F}$ using these two features.

The analysis of the representation of objects (i.e., M) involves the inspection of each token substitution set. Intuitively, as [CLS] itself has little semantics, the linguistic feature of the whose sentence should all come from each observed condition. Therefore, a better representation of the token substitution set should have a smaller standard deviation in classification scores, as all the elements in the set share the same observed condition. This is quite the case in both the pre-trained and the fine-tuned model, as we can see the better classified samples (i.e., score close to 0 or 1) have a smaller standard deviation in classification score.

Based on our intuition discussed in Section 6, we rely on the comparison of functional characterization results to evaluate the properties of morphism representation D. Specifically, in this experiment, the comparison is made between the standard deviation of classification scores across different token substitution sets output by a given representation model. Note that each morphism between two objects reflects the difference of the two corresponding observed conditions in modulating the semantics of the central tokens. In our experiment, this is manifested by the discrepancy in the stan-

dard deviation of classification scores between two samples. Interestingly, we find that samples with similar margin to the classification boundary tend to have similar standard deviation (regardless of the specific score value, in contrast to the aforementioned case of object representation). **That is to say, our experiment results show that samples with similar prediction confidence are also similar in quality of morphism representation D, which is decoupled from the specific semantics of each [CLS].**

When comparing the pre-trained to the fine-tuned setting, we can observe an interesting point for the fine-tuned model. After fine-tuning, samples near the classification boundary are associated with more significantly enlarged standard deviation in confidence score, which means these observed conditions fail to generate consistent task-related modulation to different central tokens. This is contrary of high confident samples, which shows small standard deviation along with an almost absolute prediction score. **Altogether, our results give an intriguing hint that high-confidence correct predictions tend to occur when the samples' representation of context is good in terms of both object (showing a smaller standard deviation) and morphism (consistency in standard deviation change), which validates the linguistic relevance of our propose theoretical perspective.** Note that on the y-axis, we plot the sample id, which shows no association with the mean or variance, further verifying the effect is a characteristic of [CLS] token but not any specific context.

### 7.3 EVOLUTION OF CONTEXT REPRESENTATION THROUGH A TRANSFORMER MODEL

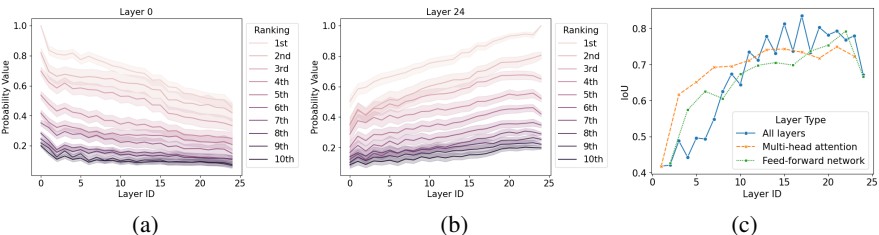

Figure 3: (a) The disappearance of neighbors in the manifold given by the first layer. (b) The emergence of neighbors in the manifold given by the last layer. The y-axis is the average probability of connection and the band width is the variance. Note that we label the embedding layer as the layer 0, the last FFN layer as the layer 24 and separately count the attention and FFN layers in one Transformer layer. Therefore, a layer with odd number index is an attention layer and a layer with even number index is an FFN layer. (c) The IoU trend between different layers. Each IoU value is calculated with last layer or last layer of the same type. The results of tokens with different POS can be found in the Appendix.

In this analysis, our focus is to characterize how the context manifold from each layer of a pre-trained Transformers model differs from each other. As this does not consider any specific downstream tasks, we use the test set of Wikitext-2 dataset, which mimics a pre-training corpus of BERT. For the visualization purpose, we sample the 10 most frequent tokens from each of the three parts of speech (i.e., noun, verb and adjective) respectively. For each token as the token of interest, we further sample 10 samples and calculate the topology around each context.

**Upper layers overwrite the context manifold topology given by lower layers.** Considering the information flow in a Transfomer-based model, each token is gradually exposed to more "integrated" information when layers move forward. When considering the set of representation vectors alone, this can be understood as the addition (with attention weights) of other token's vector to the vector of the central token. Here, we propose to analyze this change on a context-level, i.e., how the association among the context neighborhoods reflects morphisms between semantics in the discrete language space. Firstly, we examine the trajectories of evolution for the nearest neighbors appearing in the first layer (i.e., the static embedding layer) and the last layer (i.e., the output layer) (Figure 3(a) and 3(b)). This is done by calculating the average probability for the existence of an edge that connects to the real context based on how the tokens rank in the first or last layer. Clearly, we observe the gradual disappearance (loss of connection) of the neighbors in the embedding layer, as well as the emergence of new, sample-aware neighbors in the last layer. This result further validate

the relevance of "context manifold" in language modeling. The corresponding results for other layers can be found in Appendix A.1.

**Representation learning converges when the game between Multi-head attention and Feed-Forward Networks ends.** When inspecting the manifold change for specific samples, we realize an interesting balance between the attention layers and FFN layers. By statistics, we also plot the average results here (Figure 3(c)). In this experiment, we try to reveal how neighbors given by different layers are similar with each other, measured by intersection-over-union (IoU). First we have a line possessing the sequential comparison of all layers, i.e., each layer has an IoU calculated with the last layer. Interestingly, while the consistency of layers generally goes up as layers get higher, the IoUs show a trend of "zigzag" especially during the first two and upper half of layers. This result means the attention and FFN layers tend to give different results and the model tries to get an agreement between these two layer types, i.e., **the final representation of context is achieved by a co-opetition relationship of the attention and FFN modules.** Inspired by this understanding, we also calculate the IoUs solely between attention or FFN outputs (the first embedding layer serves as the starting point for both series). These results further reveal some detailed insights among the Transformer layers. At the first several layers, the results between attention and FFN layers possess a higher similarity than results between them, verifying the existence of debate. Also, as the attention layers generally have a higher consistency, a natural understanding is they tend to inject some uniform information from the whole input sequence, while the FFN layers tend to eliminate the information they don't agree (as they contain ReLU non-linearity). At the upper layers, in contrast, the overall consistency between attention and FFN layers surpasses the consistency between the layers with the same type, suggesting the discrepancy in the lower layers actually converges to a better agreement. **The above observations actually support our previous hypothesis that different layers in a model (treated as different $G$ here) learns to achieve an identity morphism in** $\mathbb{F}$, which reflects the genuine linguistic association between the probability distributions of different contexts (This result holds true for tokens from all the three parts of speech we samples as shown in the Appendix A.2). In other words, topological consistency between the attention layers and the FFN layers is a necessary condition of model convergence. In addition, we also notice that the last FFN layer seems to decrease the overall consistency a lot, which we speculate is due to the direct connection to a linear head (Ma et al., 2019).

## 8 RELATED WORK

The understanding of contextual word embedding is a field gaining more and more attention (Rogers et al., 2020; Hao et al., 2020; van Aken et al., 2019). However, it is generally regarded as a challenging task considering the complex interplay between different words. One line of researches proposes to train a probing network on the top of pre-trained word embedding, so as to find some relationship between the learned representation and the linguistic features such as syntactic tree (Hewitt & Manning, 2019; Coenen et al., 2019; Miaschi & Dell'Orletta, 2020). The process can also be realized in parameter-free fashion (Wu et al., 2020). Another line of research in this field follows the basic idea of information geometry. As each token is associated with a vector, analysis of the clustering of these vectors can generate insightful findings, such as the geometry of polysemous words (Coenen et al., 2019), isotropy in embedding space (Ethayarajh, 2019; Cai et al., 2021; Rajaee & Pilehvar, 2021). Moreover, there are works that utilize the geometric information for better model performance, possibly through refinement (Chu et al., 2019; Hasan & Curry, 2017). While our method borrows some practice from information geometry, this work is fundamentally different from the above in the way that we try to characterize the nature of contexts, while these studies try to understand how a representation vector reflects the nature of a token.

## 9 CONCLUSIONS

In this work, we propose a novel perspective to understand the contextualized word embedding, with an emphasis on the context rather than the central token. With the mathematical formulation, analysis method development, and empirical results, we hope our study can offer the community some new insights into the popular pre-trained language models.

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

# A APPENDIX

## A.1 THE OVERWRITING OF LOCAL TOPOLOGY IN DIFFERENT TRANSFORMER LAYERS

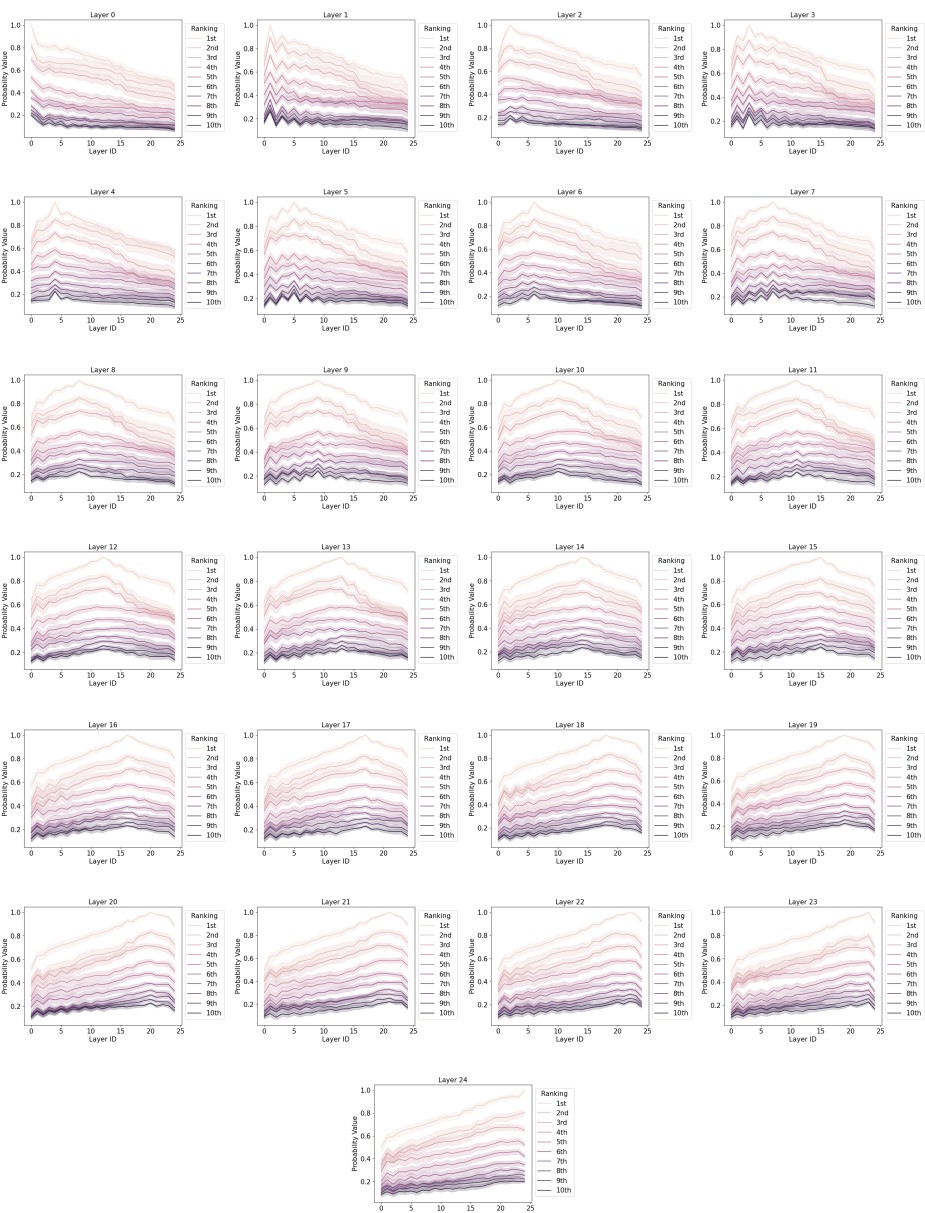

Figure 4: The overwriting of local topology in different Transformer layers. Note that we treat the embedding layer as the layer 0 and separately count the attention and FFN layers in one Transformer layer. Therefore, a layer with odd number index is an attention layer and a layer with even number index is an FFN layer. We record changing trajectory of the top 10 neighbors of the real context at each layer.

A.2    THE GAME BETWEEN THE MULTIHEAD ATTENTION LAYERS AND FFN LAYERS

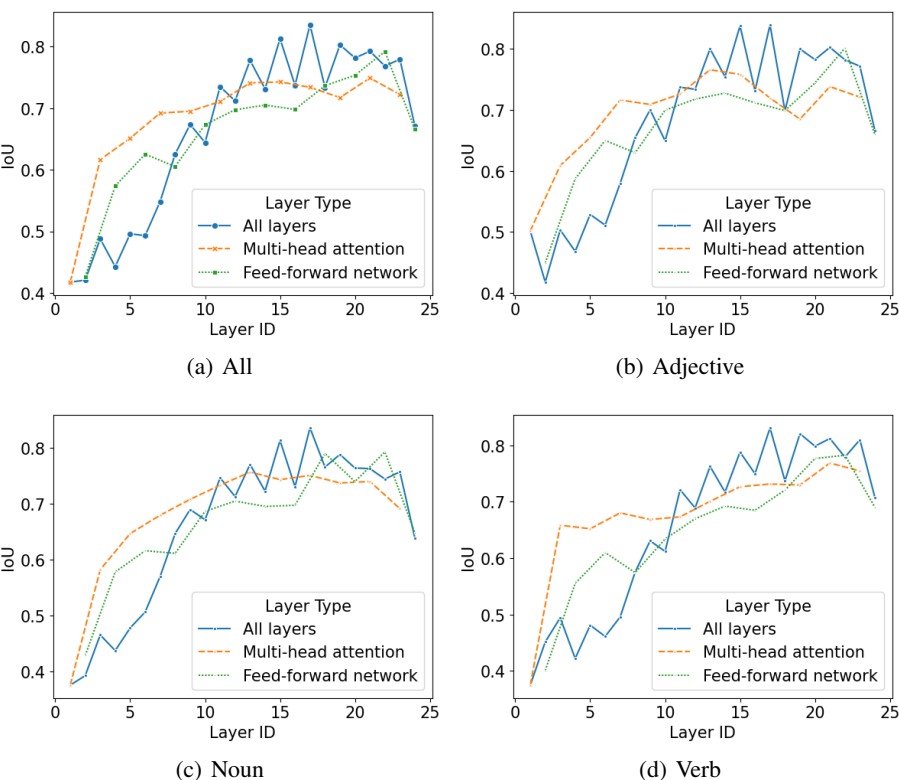

(a) All

(b) Adjective

(c) Noun

(d) Verb

Figure 5: The IoU trend results of different parts of speech. (a) All. (b) Adjective. (C) Noun. (d) Verb. We sample the top 10 most frequent tokens of each POS and get the average score.

