# OpenReview forum: "Finding One Missing Puzzle of Contextual Word Embedding: Representing Contexts as Manifold"
_ICLR.cc/2022/Conference — ICLR 2022 Submitted_

### Official Review · Reviewer_awgJ · 2021-10-16

**Correctness:** 3
**Technical Novelty And Significance:** 1
**Empirical Novelty And Significance:** 1
**Recommendation:** 1
**Confidence:** 3

**Main Review:**

This paper is an example of trying to use some complicated theories to describe some very simple observations.

This paper does not have a significant theoretical contribution because it does not have any theorem or lemma. All of the notations and definitions are just for describing their analysis method in a very abstract way. I spent quite some time understanding what those notations mean, but realize that the complex math does not help me understand the empirical findings at all, which are the main contributions of the paper.

The paper's first two empirical findings are not very interesting to me and the authors also do not explain the impact of those findings. The third finding in Figure 3 (c) could potentially deepen our understanding of BERT, but the authors need to investigate the underlying reasons to make the finding become useful.

By the way, I think the authors should also cite this work (https://aclanthology.org/2020.aacl-main.11.pdf) (and possibly other work that also tries to align the embedding space in different layers of Transformers to analyze what language model learns). These papers show that the analysis methods used in this paper are not very novel.

Minor things:
1. ELMO -> ELMo
2. Your notations are not consistent or not clearly explained. For example, What is E? Why e \in E but also C \in E? What is G_{set} = G_{m}?
3. (7) in Equation should be right justified.
4. Your Figure 1 is helpful. But if you really want to define those symbols for some reason, add more symbols into the figure would make your definitions easier to understand.
5. Your scenario A corresponds to m=n in Equation (9), right? If yes, make it more clear (if you really want to define those symbols for some reason).

**Summary Of The Paper:**

This paper reports several empirical findings.
1. Replacing the [CLS] tokens after the fine-tuning does not change the output representation a lot.
2. By using UMAP to analyze the patterns of nearest neighbors in different Transformer layers, it finds that the topological structure in layer 1 is most similar to the structure in layer 2 and less similar to that in layer 24. This shows that each Transformer layer would change the topological structure of the context embeddings.
3. Let's denote the embeddings in the Transformer in this way (H1) -> ATT -> (A1) -> FFN -> (H2) -> ATT -> (A2) -> FFN -> (H3) ..., where (Hk) is the hidden states, ATT is the multi-head attention, (Ak) is the output of the attention layer, FFN is the feed-forward network. Figure 3 (c) shows that (Hk) and (Ak) are very different when k is small, but become similar when k is large. Compared to (Hk) and (Ak), (Hk) and (Hk-1) are more similar when k is small. Similarly, (Ak) and (Ak-1) are more similar than (Hk) and (Ak) when k is small.


**Summary Of The Review:**

I vote for rejection because
1. The findings are not significant.
2. The analysis method is not novel.
3. The paper presents a few very simple experiment results using a very complicated way, which makes the paper very hard to read.

---

> ### Author Response · Authors · 2021-11-23
> **Response to Reviewer awgJ**
>
> We thank the reviewer for pointing out these points. And we apologize that some of the notations are hard to follow or contain typos. However, the key message we want to convey here is the notations we introduce are actually necessary to describe the problem we want to tackle. Please refer to the revised manuscript and the general response for a detailed description of how to connect our empirical findings to the notations (especially for the [CLS] experiment). Also, we do not agree the emphasis of this work is the empirical results. Instead, our major contribution is to introduce a novel perspective to understand how the context in a sentence modulates the semantics of the central token.
>
> For your particular question, it is true that scenario A corresponds to m=n in Eq.9 (now Eq.8). We have modified the text to make this clear. And we have also added several recommended works on language model analysis.

---

> > ### Comment · Reviewer_awgJ · 2021-11-23
> > **The rebuttal does not address my main concern**
> >
> > My main concern is about the contribution of this paper.  What kind of contributions do you want to make in this paper? What's the purpose of defining those notations?
> >
> > You say the purpose of the notations is to introduce a novel perspective. So do you want this paper to be a position paper? I never see a position paper that does not try to refute some popular beliefs in the community. What are the popular beliefs you try to refute? What can the readers learn from your arguments? I believe this paper is not a position paper because I do not see such a contribution.
> >
> > You say the purpose of the notations is to describe the problem. I still do not clearly see what the problem is and why the problem is important. People can show that the problem is important by many ways. For example, solving this problem can improve the state-of-the-art models. The existing state-of-the-art models would fail in certain kinds of cases. The problem shows that our previous understanding of the language models are wrong (e.g., the previous analysis tools have some issues). The problem has some impact to the world (e.g., deep learning hurts the environment or society). Lots of researchers are going toward a wrong direction. Could you concisely explain that what the problem is and why it is important?
> >
> > I never see a theoretical contribution without a theorem or lemma. I am not familiar with category theory, but I guess there might be some theorems that have already existed in the category theory. Do those theorems applicable here? If yes, could you explain them in the language modeling context? What do those theorems tell us? If not, what is the purpose of introducing category theory?
> >
> > Defining notations itself is a very weak contribution. The purpose of the notations is usually for describing some theorems. Since you do not introduce any theorem or lemma, I believe that you define those notations to explain your experiments. However, I can easily understand what you found in your experiments without needing those notations. The notations also do not explain why your analysis approach (UMAP) is better than the previous tools such as CCA. What's the difference between UMAP and CCA?

---

### Official Review · Reviewer_ptkS · 2021-10-29

**Correctness:** 2
**Technical Novelty And Significance:** 2
**Empirical Novelty And Significance:** 2
**Recommendation:** 3
**Confidence:** 2

**Main Review:**

Pros:
1. The attempt to explain and investigate how contexts are represented in the language model is interesting.
2. The introduction of the category theory to model the connection between contexts and their representations seems novel.

Cons:
1. The serious issue with this paper is its poor presentation. Many details are very vaguely presented which makes it very hard to understand the contribution of the paper. Firstly, as this paper studies how contexts are being represented in a language model, a clear definition of what is "context" is necessary. My guess is that the "context" of a word is all the remaining words in the sequence, but this is not clearly mentioned in the paper. If this is true, what is the difference between "context" and "observed condition $C$" in Section 3? Why not use some notations/formulas to explicitly define what is "context"? Sections 4 and 5 are very difficult to parse even after putting in lots of effort. Some important terminologies should have clear definitions (e.g., morphisms) before they can be used as the paper needs to be self-contained. Also, there should be high-level explanations/concrete examples to facilitate an easy understanding of the purpose of each definition and how they are connected to the mechanisms of contextualized representation learning. Section 7 also fails to clearly explain how the empirical findings validate the theoretical proposals.
2. The empirical results are from only one dataset representing downstream tasks (SST-2) and are largely presented in the form of case studies. More comprehensive studies are necessary to give reliable and convincing results.
3. There are a lot of typos throughout the paper; some examples:
Section 2, "understanding how the vector represent the token" -> "understanding how the vector represents the token"
Section 2, "supports our hypothesis that they both of them try to" -> "supports our hypothesis that both of them try to"
Section 3, "which motivates as to introduce" -> "which motivates us to introduce"

**Summary Of The Paper:**

This paper aims to explore how contexts are represented in a deep language model. It attempts to go beyond “token-centric” understanding in the previous contextualized representation studies and specifically tries to investigate how different contexts are being represented (e.g., when tokens are being replaced) and how the representation changes through different components (e.g., FFN and attention modules) in the language model.

**Summary Of The Review:**

While the paper attempts to study an important and interesting angle in understanding how language models represent semantics, the ideas and results are presented in a very vague manner which prevents clear understanding and judgment of the contributions in the paper. It is still unclear to me how exactly are "context" represented in language models and what insights can be obtained from the paper, even after reading the paper several times.

---

> ### Author Response · Authors · 2021-11-23
> **Response to Reviewer ptkS**
>
> We thank the reviewer for the comment. In the revised manuscript, we have improved our presentation in motivation, notation, formulation, and experiment descriptions. Please also refer to the general response for a concrete discussion on the key points. We hope the revised version can convey our message correctly.
>
> Due to the time limit, we cannot systematically revise our experiments. Nonetheless, through the current experiments, we still show how the pre-trained language model commonly used today can represent at least some of the semantic effects of different contexts with our theoretical framework. We will continue to improve the experimental efforts in the future.

---

> > ### Comment · Reviewer_ptkS · 2021-11-30
> > **Thanks for the rebuttal**
> >
> > I would like to thank the authors for the efforts put into the rebuttal. I agree with the other reviewers that the paper still needs to be improved in clarity in order to be considered for publication. For example, start from basic/widely-used definitions of concepts (_e.g._, defining tokens, contexts with rigorous notations) and then introduce the new/advanced concepts (_e.g._, category theory). This will give a good general picture of the framework. Also, try to make your contribution more concrete (instead of making a very big and vague claim); it could be analytical, theoretical, or empirical, but it has to be made clear to the readers. I keep my original rating but also would like to encourage the authors to continue this interesting direction and spend more effort improving the manuscript.

---

### Official Review · Reviewer_K1ke · 2021-10-30

**Correctness:** 2
**Technical Novelty And Significance:** 2
**Empirical Novelty And Significance:** 2
**Recommendation:** 3
**Confidence:** 4

**Main Review:**

This paper tries to build a theory about how context is represented in pretrained language models. This is indeed a meaningful direction as current literature, although extensive, does not contain targeted studies about context. However, this paper is severely flawed in the following aspects:

## Basics of probability and pretrained LMs

Equation 1:

- a masked language model outputs the probability of p(masked word | all other tokens in the sentence), NOT p(context |  C, w).
- what does the word *context* mean in eq.1? It seems to be a random variable since it has a probability. So is it a discrete random variable, or a continuous random variable, or a sequence/set of discrete/continuous random variables?
- what does *C* mean? The paper says it is the observed unmasked tokens but isn't this part of the context? My understanding is that the *context* of a given masked word is all other words within the sentence (either other words are masked or not, as recall 15% words are masked in BERT).
- what does the symbol ~ mean in Eq.1? Does it read "follows this distribution"? If it is, does it induce that the symbol G(C, w) is a random variable? The paper says G is "a contextual language representation", but what is it? Is it a sequence of output vectors of the last layer of BERT? If they are, they are deterministic (not random variables), so how can they have distribution?

The term "observed condition C"

- If it means "observed unmasked tokens", why re-name it as "observed condition", since the former is apparently more understandable?

Equation 2:

- Again, what does the symbol ~ read? Does it mean "follows the distribution of"? If yes why the right hand side is a set, but not a probability distribution?

A basic rule about writing equations and symbols is to always define them properly before using them. In this case, I strongly encourage the authors to discuss the definitions of symbols in their equations. In addition to the above questions, the authors may also want to discuss

- G(C, w), is this a function G applied to variable C and w? If yes what is the domain and what is the image?
- Similarly, for P(C), is P a function applied to a variable C? is C continuous or discrete? is C a scalar or a vector?

## The definitions of symbols are unclear and without definition or explanation, equations are not explained

There are a lot of concepts directly used without explanation or reference to the background. To name a few:

- Definition 4.1, what is the use of the sigma-filed E? Is it the vocabulary? If yes why not directly say vocabulary?
- Definition 4.2, what is X? Is it a word at a certain location? If yes why not directly say it is a word at a certain location? Does a context mean a set of functions takes a sentence (excepted a mask word) as the input, and output a distribution over the vocabulary (because the paper says "A context is ... a set ... each element can be expressed as a ... function")?
- If a context C is a set of functions, as is in definition 4.2, why the symbol C \in E in equation 3? Isn't E the vocabulary?
- Why introduce all these new symbols and terminologies (like "observation condition") without explanation, when we already have perfectly understandable terms (like "words" and "sentences")?
- For definitions 4.3 and 4.4, I get more lost as these are longer equations without definition and explanation. It is possible that it is only me that have difficulty reading these equations, as I am not so familiar with category theory (but I do know combinatorial categorial grammars well, more on this later), but I find it hard to read equations from 5 to 9. There are a lot of ambiguities when I try to understand, for example:
- It seems that "the representation of context" and "context" are two different things, as this paper writes "the representation of context as a functor". Yet with pretrained language models, the word "representation" usually refers to vectors, so how do these two compare?
- Combining definition 4.2, equation 5, and equation 11, I guess the paper is trying to say: the conditional probability of a masked word (def. 4.2) is the same (equality 5) in different pretrained language models (Gm and Gn in Eq.11) if the context is the same. Is this how equation 11 should be interpreted? If yes, does this paper consider different training datasets of different pretrained LMs?

## Lack of connections between the category theory and the language model

Is it possible that this paper gives a one-to-one mapping between all introduced concepts to standard terminologies in NLP? For example:

- What does the term "category" correspond to? A context?
- What does the term "object"  correspond to? A context in a specific sentence?
- What does the term "morphism" correspond to? A mapping between different contexts?

## Important related work is missed

How does the introduced theory compare to classical combinatorial categorial grammars? What if we build a supervised/ unsupervised CCG tagger upon a pretrained language model like Liu et. al. 2019 (In this case each word is also associated with a linguistic category)?

## Insufficient experiments

Generally, the current experiments are too simple and not well-explained. Specifically,

- for figure 2a, my understanding is that the authors substitute the original token to another token, then use UMAP to get the probability of the edge, but how does this compare to the original masked LM where one can directly get a distribution of possible substitution words? Why bother with UMAP and what does it tell us?
- What new things can we learn from figure 2bc? My understanding of them is the classes are more separated after fine-tuning (if I have not missed anything), but isn't this trivial?

## References

- Nelson F. Liu, Matt Gardner, Yonatan Belinkov, Matthew E. Peters, Noah A. Smith. Linguistic Knowledge and Transferability of Contextual Representations. NAACL 2019

**Summary Of The Paper:**

This paper aims for providing a theory for understanding the context representation in pretrained language models. It presents a category theory then some analysis of representations from pretrained LMs. This inspiration is good.

However, this paper is severely flawed in multiple aspects: the basic notations and understanding of LMs seem to be wrong; the definitions of concepts are unclear and ambiguous; many symbols are used without definition or explanation; equations are not explained; there is barely any connections between the category theory and the language model; important related work is missed, and the experiments are quite simple and insufficient. I think this paper needs significant improvements in terms of basic understanding, writing, explanation, and experiments.

**Summary Of The Review:**

Although the goal of this paper (to analyze the effects of context) is good, it suffers from severe issues as mentioned above. I would recommend a re-writing, especially the basics (like what kind of variable a context is), with more motivations, explanations, and sufficient experiments.

---

> ### Author Response · Authors · 2021-11-23
> **Response to Reviewer K1ke**
>
> We thank the reviewer for the comments. We will post our response below.
>
> 1.	Basics of probability and pre-trained LMs
>
> Following the distributional hypothesis, the modeling target of a representational model is p(context |central token). In this sense, “context” is a random variable. Note that this is different from the loss function of an MLM task. In an MLM training, one needs to add a linear head on the top of representation model to fit the loss function. We have systematically revised our definition and notations in this part. Please refer to the general response and Section 3 of the main text.
>
> 2.	Formulation of context representation and connection to LM
>
> The introduction of sigma-field is to facilitate the definition of context probability density function. The reviewer’s other concerns should be addressed by our revised Section 4 and our discussion in the general response.
>
> 3.	Related work
>
> After reading the suggested reference (Liu et al., 2019), we find it hard to associate our understanding to combinatorial categorial grammar.
>
> 4.	Experiment
>
> For Figure 2(a), the reviewer is right that we can also use the linear head from the original MLM task. In fact, this falls into the functional characterization as we proposed in Section 6. The introduction of UMAP aims to characterize another aspect of category M, i.e., the topology between different elements. It tells are which elements are closer in the manifold, which cannot be addressed by a score given by a linear head.
>
> For Figures 2(b) and (c), we have revised the corresponding part of the main text and general response (3/3). Generally speaking, the aim of our experiment is not to characterize the classification scores but how well different observed conditions modulate the semantics of [CLS].

---

> ### Comment · Reviewer_K1ke · 2021-11-29
> **Response to rebuttal**
>
> I thank the authors for their response. However, I still find it hard to increase my scores due to:
>
> * Unclarity of the basic notation and concepts, as is also discussed by other reviewers.
> * Insufficient experiments.
>
> Of course, it could be that it is me that do not have the right mindset to understand the contribution. However, I do believe there are a set of basic rules when introducing new theories. Importantly one need to
>
> * Clearly explain the basic notations and concepts, using standard math languages. For example, when saying "context" is a random variable, is it a discrete r.v.? Or continuous r.v.? Is it integer valued, or real valued, or a vector of integers / real values? or a set of integers / real values?
> * Follow the recognized convention. For example, the community tend to view the last layer of BERT as a continuous vector (not a random variable and nor a distribution).
> * Introduce new concepts slowly, step by step, with illustrative examples. Be specific and concrete, instead of being abstract.
>
> I would like to suggest the authors to put efforts on the basics and simple intuitive examples in their following modifications. Try to decrease the level of abstraction and increase the portion of tangible examples. After all, appreciation should be built on reviewers' basic understandings of the paper.

---

### Official Review · Reviewer_3Rbc · 2021-11-02

**Correctness:** 2
**Technical Novelty And Significance:** 2
**Empirical Novelty And Significance:** 2
**Recommendation:** 3
**Confidence:** 3

**Main Review:**

This paper intends to understand “how does a language representation model represent contexts” which is different from a more popular direction of understanding “how does the context affect the representation of the tokens”. The paper formulates the contextualized language representation problem using category theory. Authors claim that previous works that focus on substituting tokens by keeping the context fixed miss a lot of useful information. According to their proposed framework, representation of the context contains information about the representation of tokens that get substituted while keeping the surrounding conditions unchanged as well as information about the transitions (morphisms) in the representations as a result of these substitutions. However, quantifying the above information is challenging, and this paper intends to analyze this information from a functional (task-related projection) and topological (UMAP) perspective.

Before you read further, it is important to mention that I do not have a background in category theory or Manifold learning so I have judged this paper purely on the basis of convincingness of the arguments, quality of experiments, and interpretation of the results.

Overall, I think that paper identifies a very niche problem and provides interesting ideas about how we can think about representation of the contexts in language modelling. But in the same light, I do not think that the paper is well-written to be published at this conference. The paper needs serious revamping in terms of story, writing style and presentation.

In terms of the story:

- The main takeaway of the paper remains unclear to me for most of the paper as the authors were not precise with communicating their intentions. For instance, the paper’s abstract and various other parts indicate that the work would explain “how a model represents context itself,” whereas, in the motivation, the paper states that the authors are trying to tackle “how a representation model learns to represent contexts”. I believe that these two statements are not identical. The latter claim is about how context emerges in the model when learning happens during training. In contrast, the former is about analyzing how the context is represented once the learning is complete. The authors should state both of these clearly if they are trying to answer both the questions.

- In the abstract, the paper claims that the work would shed light on the improvements in Transformer-based language representation models. However, the paper does not mention how it plans to do so in the main text. Additionally, I do not understand the nature of improvements the authors are referring to.

In terms of the writing style, there are several places where the authors need to rephrase sentences or correct grammatical errors. To list some:
- In early days, word embeddings ~~are~~ were static, i.e., the representation is was solely determined by a word’s identity…
- GPT ~~revolutionize~~ revolutionized this technique
- When compared to static embedding, ~~mechanistic~~  (use a better word here?) studies of the contextual embedding generally lag behind.
- They also point out **that** the embedding space is generally anisotropic. Also what do you mean by anisotropic? A reader without a manifold learning background is likely to phase out after coming across such terms if they are not explained succinctly.
- On the other hand, several works try to uncover how contexts influence the geometry of embedding space. (Missing references)
- Therefore, very few studies touch (touch upon?) another critical problem, i.e.,.
- untouched fundamental problem in deep language learning. I am reading the term **deep language learning** for the first time. Can you please clarify it a bit more?
- The ~~fine-turning~~ fine-tuning process …
- FFN layers identifies a ~~Game~~ game between…
- hypothesis that they both of them (not sure what you meant here) try to fit an identity morphism that reflects bona fide linguistic associations.
- I believe that the readers will benefit from detailed explanations of the terms presented in the mathematical equations. For instance, equations (6)-(11) have been introduced together in Definition 4.4 without much explanation as to what each of it intends to convey.

In terms of the presentation:
- It seems as if Figure 1 is an important cue for the readers to understand their framework but it lacks descriptive captions or a reference in the main text that could explain the idea succinctly unlike Figure 2 and Figure 3.
- The graphs in Figure 3 should somehow indicate that if Layer i is Feed-forward network (FFN) then Layer (i+1) is Multi-head attention. Make it clear in the caption as well.

The authors test their method on just one of the GLUE classification dataset, SST-2, using Wikitext-2 as a pre-training corpus. I believe that the authors should include more datasets and discussion to make the experiment results more convincing. There are other single sentence classification datasets in GLUE such as CoLA, and other inference based classification tasks such as MNLI (https://openreview.net/pdf?id=rJ4km2R5t7).

After multiple parses through your experiments section, I am unclear if the authors perform token substitution around [CLS] token or they replace [CLS] token itself. If I think about this experiment from Scenario A’s (Section 5.2) and Figure 2 perspective, it suggests that the authors intend to convey the former. However, the authors mention  “when replacing the [CLS] tokens with other tokens, the representation of the whole sentence does not change a lot” which aligns with the later understanding. The authors should clarify what they actually intended to do in this experiment. The results from the experiment suggest that finetuning helps in learning new patterns and pre-training patterns are forgotten which is not that surprising based on our current understanding of fine tuning.

I found the authors’ observations on the conflict between Multi-head attention and Feedforward network quite interesting. I believe that the authors should analyze this result and conduct more experiments around this idea on more datasets. It would be interesting to see if the same observation holds for other pretraining corpuses as well which extend beyond English as well. It might be even the case that the observations the authors made were a result of the quirks (linguistic properties) of English rather than something being innate to attention-based language modeling.

**Summary Of The Paper:**

This paper intends to understand “how does a language representation model represent contexts” which is different from a more popular direction of understanding “how does the context affect the representation of the tokens”. The paper formulates the contextualized language representation problem using category theory. Authors claim that previous works that focus on substituting tokens by keeping the context fixed miss a lot of useful information. According to their proposed framework, representation of the context contains information about the representation of tokens that get substituted while keeping the surrounding conditions unchanged as well as information about the transitions (morphisms) in the representations as a result of these substitutions. However, quantifying the above information is challenging, and this paper intends to analyze this information from a functional (task-related projection) and topological (UMAP) perspective.

**Summary Of The Review:**

It is important to mention that I do not have a background in category theory or manifold learning so I have judged this paper purely on the basis of convincingness of the arguments, quality of experiments, and interpretation of the results.
Overall, I think that paper identifies a very niche problem and provides interesting ideas about how we can think about representation of the contexts in language modelling. But in the same light, I do not think that the paper is well-written and rigorous enough to be published at this conference. The paper needs serious revamping in terms of story, writing style and presentation. I provide detailed explanation of my concerns in the main review.

---

> ### Author Response · Authors · 2021-11-23
> **Response to Reviewer 3Rbc**
>
> We thank the reviewer for the helpful comments. The general response posted has included most of the clarification of our motivation, method and experiment, and we think the current content has been greatly improved in communication. The following listed the response to the reviewer.
>
> 1.	Main takeaway:
>
> Thanks for pointing this out. Our purpose is to explain how a model represents context, in terms of how it modulates the semantics of the central token. Our scope does not involve learning.
>
> 2.	Further improvements of PLM based on our understanding
>
> We have removed this part from our abstract as this is not the focus of our work.
>
> 3.	Writing style and presentation:
>
> We have revised the manuscript as proposed by the reviewer. We are very grateful for these suggestions. In particular, the figure presentation and captions are improved.
>
> 4.	Experiments:
>
> Due to the time limit, we cannot systematically revise our experiments. Nonetheless, through the current experiments, we still show how the pre-trained language model commonly used today can represent at least some of the semantic effects of different contexts with our theoretical framework. We will continue to improve the experimental efforts in the future.

---

> > ### Comment · Reviewer_3Rbc · 2021-11-29
> > **Response to Rebuttal**
> >
> > Thanks to the authors for their efforts during the rebuttal period. I really appreciate the fact that some of the low-level changes were taken care of in the manuscript. However, I still don't fully understand the concept of context and how it is different from the observed condition. It is your preliminary section which causes a lot of confusion. You can not do anything now but I would highly suggest the authors to give some toy example in form of sentence which adds graphics to their equation (1). Sorry but I don't understand your improvements in the category theory notations as I don't have the requisite background. It would be great if you can conduct broader set of experiments so that your findings get more weight in future submissions.

---

### Official Review · Reviewer_BMjD · 2021-11-02

**Correctness:** 3
**Technical Novelty And Significance:** 2
**Empirical Novelty And Significance:** 2
**Recommendation:** 3
**Confidence:** 2

**Details Of Ethics Concerns:**

This paper analyzes pretrained language models, which have been shown to potentially have biases, but the methods used are not of ethical concern.

**Main Review:**

Strengths:
- The proposed approach is interesting and suggests an alternative viewpoint to understanding contextualization in language models.

Weaknesses:
- The empirical results do not seem to rely substantially on the methodological contributions. In particular, experiments are performed with local probing (UMAP) over a small number of neighbors (15) to determine manifolds compared with a general functor over manifolds presented in Sections 4-6. This form of analysis is quite similar to that performed in Cai et al. 2021, except over contexts rather than tokens. Additionally, the layerwise analysis is quite similar to that performed by Aken et al. 2019 CIKM in the context of question answering.
- The paper is a bit hard to follow, especially the description of input and output categories in Section 4. Figure 1 is helpful for this.
- The experiments are somewhat limited and, as mentioned previously, similar to those previously conducted in prior works except over contexts rather than the central word.
- For the [CLS] experiments, it seems straightforward that the functional characterization is mixed before fine-tuning and concentrated after finetuning as the [CLS] token is rarely used in masked language modeling (at least for the BERT formulation of this objective).


**Summary Of The Paper:**

This paper proposes a framework, using concepts from category theory, to analyze the mechanisms through which Transformer based language models perform contextualize representations. In contrast to prior work, which analyzes contextualization of representations through “token-centric” probing, the authors first cast the language representation problem as a functor from an input category (encoding of observed context) to an output category (d-dimensional space). Next, a manifold learning approach is analyzed in the case of a fixed representation model and treating the outputs at each layer as a different representer. Lastly, the authors perform a range of experiments to better understand various aspects in transformer models including how contexts are represented for [CLS] and the change in context representations by layer.

**Summary Of The Review:**

Approach for modeling contextualization in language models using tools from category theory and manifold learning. Interesting approach but there is a gap between the methodology and the experimentation. Additionally, experimentation is similar to studies conducted previously with a focus on substitutions in context rather than center token.

---

> ### Author Response · Authors · 2021-11-23
> **Response to Reviewer BMjD**
>
> Thanks for your valuable suggestions. Our response is listed below. Please note that we have revised the methodology as well as the experiment part to provide a clearer presentation and stronger connection between them. The reviewer can also refer to the general response for the key points.
>
> 1.	Comparison with previous works
>
> The manifold learning in (Cai et al. 2021) involves representations of tokens from different species and different samples, while we propose to learn a collection of manifolds, each of which consists of token embeddings with the same observed condition but different central tokens. Note that our work does not aim to propose a novel manifold learning method. Therefore, the key difference lies in the definition of manifold and the linguistic meaning behind it. Our proposal to use token substituted set to characterize the semantic modulation effect of the observed condition is original and non-trivial.
>
> As mentioned by the reviewer, Aken et al. 2019 tries to design some general and QA-specific prosing tasks to understand the function of each layer in BERT. It tries to relate each layer to some function (e.g., relation classification or question-fact matching), so the clustering is done for the representation of words that appear in the text. In comparison, we try to define a metric to represent how an observed condition affects the central token (not to attribute this effect to any manually designed task), and the manifold learning is conducted with the token substitution set. This comparison actually brings an interesting point. At a higher level, their work tries to describe the effect of the contextual information to each appearing central words directly (through clustering and human definition), while our work tries to disentangle the effect from the central token and its context, and the emphasis of our method is the latter. Therefore, these two kinds of works are actually complementary.
>
> 2.	Description of input and out of categories
>
> We have updated the description in Section 4. The motivation is also elaborated in the general response as well as in the main text.
>
> 3.	The [CLS] experiments
>
> We have revised the corresponding part of the main text, which can also be found in General Response (3/3). Generally speaking, the aim of our experiment is not to characterize the classification scores but how well different observed conditions modulate the semantics of [CLS]. So the parsing of the experiment results is closely related to the understanding of our perspective defined in the method part.

---

> ### Comment · Reviewer_BMjD · 2021-11-29
> **Response to Rebuttal**
>
> I appreciate the authors efforts in their response; however, I am in agreement with awgJ and find the contributions, which consist primarily of notational constructs, to be neither meaningful or particularly novel. I recommend that the authors either focus on making + proving coherent theoretical claims or providing novel empirical analysis which provides a stronger understanding of language models.

---

### Author Response · Authors · 2021-11-09
**General response to all reviewers (1/3)**

Thank you very much for reviewing our manuscript. We understand that reading such content with a lot of abstraction and mathematical formulation needs a lot of labor and patience. At this stage, we hope this rebuttal can serve as a valuable open communication with the community and try to clarify the message we convey.  Here we try to address the general concerns of the reviewers.

[Definition of context and observed condition]

One particular difficulty in understanding our idea comes from multiple roles of “context” in contextual language representation. Here we explain in detail the meaning of each related term in the manuscript.

Following the distributional hypothesis[1], the output of a representational model represents the probability of the appearance of certain tokens (i.e., context) around a given token (i.e., central token), i.e., p(context|w). Here, “context” is a random variable.

In contextualized language modeling[2], the distribution of context is conditioned not only on the central token but also on what really appears around the central token in each sample. In this sense, we introduce the term “observed condition C” to denote the observed context. That is to say, observed condition C corresponds to an actual value of random variable context given a specific sample, which makes the modeling target be p(context|C, w).

Using the formal language of probability theory[3], in BERT-style language model, we use a three-dimensional tensor (token species, position and type) to define the sample space and take the sigma-field of sample space as the event space (Definition 4.1). Note that to describe the collection of probability distribution of different central tokens, we introduce a discrete random variable set K={ki} and the distribution of each ki’s image corresponds to the distribution of context given a certain central token i (Note that i is also defined as a triplet of species, position and type).

[Representation of context: Why and What]

During the development of distributed language representation, we have witnessed the transition from static representation to contextualized representation (see above). The core idea of introducing the latter method is to introduce the impact of surrounding tokens to the central token. Intuitively, the effect of different context conditions (termed as observed condition C in our manuscript) should be semantically related, which is, however, not explicitly modeled in the current representation scheme. And how to design a representation scheme of context to reflect the nature of this kind of effect becomes an interesting question.

Ideally, such a representation scheme should have the following two features. Firstly, it should be compatible with the word representation mentioned above. Secondly, the effect of the central token and observed condition should be disentangled, which makes the latter focusing on how different they are in affecting the central token’s semantics.

Reference

[1] Harris, Zellig S. "Distributional structure." Word 10.2-3 (1954): 146-162.

[2] Peters, Matthew E., et al. "Deep contextualized word representations." Proceedings of NAACL-HLT. 2018.

[3] Stroock, Daniel W. Probability theory: an analytic view. Cambridge university press, 2010.

---

### Author Response · Authors · 2021-11-23
**General response to all reviewers (2/3)**

[Rationale of introducing category theory]

To achieve the goal stated above, we design the following representational scheme. Firstly, for each observed condition, we generate a collection of probabilistic density functions F(C) by putting in all possible central tokens (Definition 4.2), so each element corresponds to a contextualized representation. This kind of collection reflects how each observed condition affects the semantics of the different central tokens, and comparing different collections reflect how different contexts affect the semantics. On the other hand, as mentioned above, to disentangle the effect of observed condition and central token, we study the nature of the collection, other than its elements.

In other words, we expect that two disentangled aspects of nature should be considered in the representation of observed conditions.
1)	For each observed condition, how it can affect the semantics of the central token.
2)	How different observed conditions modulate the semantics differently.

The first point can be readily addressed by characterizing the element in the aforementioned collection. However, the second point cannot be characterized when we merely take the collection as a set (termed as “token substitution set” in our manuscript). Therefore, we introduce some concepts from category theory to formulate the representation of context (Definition 4.3). In particular, the concept “morphism” characterizes the mappings between different token substitution sets (corresponding to the semantic effects of different observed conditions).

The rest of the mathematical definition tries to establish the connection between the current language representation model (e.g., BERT) and the representation of context. Note that each observed condition is converted to a token substitution set, each element of which can be represented by a vector output by a language representation model. Following this design, we also introduce the concepts from category theory to formulate the representation of vectors (Definition 4.4).

[Connection between the theoretical proposal and empirical results]

Our theoretical framework proposes a way of representing contexts regarding how they modulate the semantics of the central token. And the experiment part aims to validate the proposed representation scheme makes sense in language modeling.

It is noted that the reviewers find it hard to connect our theoretical proposal to the experiments on [CLS], which we will discuss in detail below. As stated above, we define the representation of each observed condition around [CLS] as a collection of word representations of the token substitution set. Moreover, the linear head projects the whole set to different real number classification scores. Our goal is to understand how each observed condition modulates the semantics of [CLS] differently, and the evaluation of this modulation should be disentangled from specific semantics as far as possible (as discussed in the motivation part).

Figures 2(b) and 2(c) show the results from both finetuned and pertained models, respectively. Note that each dot shows two features of a sample, i.e., the classification score of the original sentence beginning with [CLS] and the variance of classification scores of all the elements in the token substitution set. In particular, the classification score indicates the task-related (a binary sentiment analysis) semantics of the sample. And the variance represents the task-related magnitude of semantic shifts across the token substitution set, which simplifies the comparison between different sets to the comparison of different variance values. We then show how we analyze the representation of objects and morphisms of context category F using these two features.

---

### Author Response · Authors · 2021-11-23
**General response to all reviewers (3/3)**

The analysis of the representation of objects (i.e., M) involves the inspection of each token substitution set. Intuitively, as [CLS] itself has little semantics, the linguistic feature of the whose sentence should all come from each observed condition. Therefore, a better representation of the token substitution set should have a smaller variance in classification scores, as all the elements in the set share the same observed condition. This is quite the case in both the pre-trained and the finetuned model, as we can see the better classified samples (i.e., score close to 0 or 1) have a smaller variance in classification score.

Based on our intuition discussed in Section 6, we rely on the comparison of functional characterization results to evaluate the properties of morphism representation D. Specifically, in this experiment, the comparison is made between the variance of classification scores across different token substitution sets output by a given representation model. Note that each morphism between two objects reflects the difference of the two corresponding observed conditions in modulating the semantics of the central tokens. In our experiment, this is manifested by the discrepancy in the variance of classification scores between two samples. Interestingly, we find that samples with similar margin to the classification boundary tend to have similar variance (regardless of the specific score value, in contrast to the aforementioned case of object representation). That is to say, our experiment results show that samples with similar prediction confidence are also similar in quality of morphism representation D, which is decoupled from the specific semantics of each [CLS].

When comparing the pre-trained to the finetuned setting, we can observe an interesting point for the finetuned model. After finetuning, samples near the classification boundary are associated with more significantly enlarged variance in confidence score, which means these observed conditions fail to generate consistent task-related modulation to different central tokens. This is contrary of high confident samples, which shows small variance along with an almost absolute prediction score. Altogether, our results give an intriguing hint that high-confidence correct predictions tend to occur when the sample’s representation of context is good in terms of both object (showing a smaller variance) and morphism (consistency in variance change), which validates the linguistic relevance of our propose theoretical perspective.

Admittedly, the empirical results are not the emphasis of this manuscript, and we take a lot of simplification in the experiment. For example, the above comparison in terms of variance ignores the complex relationship among different elements in the token substitution set (The next experiment on Transformer layers somehow covers that part). Nonetheless, through these experiments, we still show how the pre-trained language model commonly used today can represent at least some of the semantic effects of different contexts with our theoretical framework. We will continue to improve the experimental efforts in the future.

[Miscellaneous]
We thank the reviewers for pointing out several miscommunications and typos. We will also revise the mathematical formulas to fix some mistakes in the new version.

---

### Decision · Program_Chairs · 2022-01-20

**Decision:**

Reject

**Comment:**

This paper proposes a theory for understanding the context representation in pretrained language models. The strengths of the paper, as identified by reviewers, are in the importance of an attempt to explain contextualization in language models, and in the novelty of using the category theory to model the connection between contexts and their representations. However, all the reviewers identify several major weaknesses, including flawed/incoherent definitions of concepts in the proposed theory and insufficient experimental results. Although the authors' rebuttal put a great deal of effort to address raised concerns, all five reviewers agree (and provide very detailed justifications along with suggestions for improvements) that the work is not yet ready for publication.